# Non-Contrast-Enhanced and Contrast-Enhanced Magnetic Resonance Angiography in Living Donor Liver Vascular Anatomy

**DOI:** 10.3390/diagnostics12020498

**Published:** 2022-02-15

**Authors:** Chien-Chang Liao, Meng-Hsiang Chen, Chun-Yen Yu, Leung-Chit Leo Tsang, Chao-Long Chen, Hsien-Wen Hsu, Wei-Xiong Lim, Yi-Hsuan Chuang, Po-Hsun Huang, Yu-Fan Cheng, Hsin-You Ou

**Affiliations:** 1Department of Diagnostic Radiology, Kaohsiung Chang Gung Memorial Hospital, Kaohsiung 833401, Taiwan; liao1009@gmail.com (C.-C.L.); sperfect@msn.com (M.-H.C.); y7192215@ms17.hinet.net (C.-Y.Y.); leolctsang@gmail.com (L.-C.L.T.); lordblue607@yahoo.com.tw (H.-W.H.); rahxaphon01@gmail.com (W.-X.L.); jenaqwer@gmail.com (Y.-H.C.); qwe79103@gmail.com (P.-H.H.); 2Department of Surgery, Kaohsiung Chang Gung Memorial Hospital, Kaohsiung 833401, Taiwan; clchen@adm.cgmh.org.tw

**Keywords:** contrast agent, hepatic anatomy, inflow-sensitive inversion recovery, liver transplant, living donor, signal intensity

## Abstract

*Background:* Since the advent of a new generation of inflow-sensitive inversion recovery (IFIR) technology, three-dimensional non-contrast-enhanced magnetic resonance angiography is being used to obtain hepatic vessel images without applying gadolinium contrast agent. The purpose of this study was to explore the diagnostic efficacy of non-contrast-enhanced magnetic resonance angiography (non-CE MRA), contrast-enhanced magnetic resonance angiography (CMRA), and computed tomography angiography (CTA) in the preoperative evaluation of living liver donors. *Methods:* A total of 43 liver donor candidates who were evaluated for living donor liver transplantation completed examinations. Donors’ age, gender, renal function (eGFR), and previous CTA and imaging were recorded before non-CE MRA and CMRA. CTA images were used as the standard. *Results:* Five different classifications of hepatic artery patterns (types I, III, V, VI, VIII) and three different classifications of portal vein patterns (types I, II, and III) were identified among 43 candidates. The pretransplant vascular anatomy was well identified using combined non-CE MRA and CMRA of hepatic arteries (100%), PVs (98%), and hepatic veins (100%) compared with CTA images. Non-CE MRA images had significantly stronger contrast signal intensity of portal veins (*p* < 0.01) and hepatic veins (*p* < 0.01) than CMRA. No differences were found in signal intensity of the hepatic artery between non-CE MRA and CMRA. *Conclusion:* Combined non-CE MRA and CMRA demonstrate comparable diagnostic ability to CTA and provide enhanced biliary anatomy information that assures optimum donor safety.

## 1. Introduction

Living donor liver transplantation (LDLT) is becoming more common in response to the shortage of cadaveric livers and has advantages such as minimal ischemic time, allowing it to be performed at a scheduled time after detailed preoperative evaluation; however, living donor safety remains an important issue in this procedure [1,2].

Hepatic vascularity is known for substantial anatomic variation between individuals, and detailed imaging evaluation plays an essential role in reducing transplant complications and ensuring donor safety [3]. Computed tomographic angiography (CTA) is a rapid and accurate method by which to delineate not only vascular anatomy but also liver parenchymal lesions [4]. It replaces conventional digital subtraction angiography, which requires an invasive procedure and is no longer performed routinely, making it a reliable imaging modality for vascular anatomy surveys. Contrast-enhanced MR angiography (CMRA) has also been applied in the preoperative evaluation of LDLT. Both CTA and CMRA demonstrate agreement and complete information regarding the hepatic vasculature [5]. The advantages of MRI study include that it is free of ionizing radiation and delineates detailed bile duct anatomy [6,7].

However, both CTA and CMRA require the injection of intravenous contrast medium, which is noted for risk of allergic reactions and nephrotoxicity [8,9], especially for gadolinium-based contrast medium. Risks include not only nephrogenic system fibrosis in patients with impaired renal function, which still lacks optimal treatment, but long-term safety such as brain deposits of gadolinium must be considered [10].

Non-contrast-enhanced MR angiography (non-CE MRA) angiography is a novel technique for imaging vascular anatomy without injection of contrast medium, in which the signal for each vessel is based on the associated blood flow. This technique has been widely used in imaging intracranial arteries. By overcoming the respiration-related motion artifacts, non-CE MRA has also demonstrated reliable diagnostic value in abdominal vascular imaging and renal transplantation evaluation [11].

Multiple non-CE MRA sequences have been developed, such as time-of-flight or phase contrast techniques [12,13], which have been commonly used in intracranial arteries and cardiovascular system imaging but have limitations in the hepatic vascular evaluation due to the overlapping hepatic arteries, portal veins, and hepatic veins [14]. Other non-CE MRA techniques such as fast advanced spin echo (FASE) and true steady-state free precession (SSFP) acquire vascular images based on the T2/T1 radio, which rapidly produces high signal intensity in blood and delineates vascular structure with an optimal signal-to-noise ratio (SNR) [15].

The three-dimensional (3D) non-CE MRA is acquired by IFIR (inflow-sensitive inversion recovery) by choosing an optimal inversion time and proper positioning of the tagging pulse to suppress unwanted blood flow signal; as such, this sequence delineates the hepatic arteries, portal veins, and hepatic veins separately without overlapping, which allows clearer interpretation of each vessel [16]. In addition, by using the non-CE MRA technique, particularly when other imaging studies have failed to provide detailed vascular anatomy, the technique can even be repeated and will still reduce contrast medium injections.

Donor vascular anatomy evaluation remains an important issue in successful living donor liver transplantation. Surgeons must be aware of the detailed vascular structure to select the optimal graft and prevent vascular complications in the donor [17]. The objective of this study was to evaluate the diagnostic performance of non-CE MRA, CMRA, and combined non-CE MRA plus CMRA in liver vascular anatomy and to compare these with CTA performance.

## 2. Materials and Methods

### 2.1. Donor Candidates

A total of 44 liver donor candidates were enrolled between June 2014 and January 2016, including 22 females and 22 males with mean age 32 years (range, 18–55 years). Forty-three donor candidates successfully completed the extra non-CE MRA examination, and one donor failed to gain optimal image due to a respiratory gating problem. All donors involved in this prospective study underwent routine CTA and MRI studies for pre-transplantation evaluation. The study protocol was approved by the ethics committee of Chang Gung Memorial Hospital (IRB 104-7097B), and all donor candidates were informed of the transplantation process and provided signed informed consent.

### 2.2. CTA Examination

All CTA studies were performed using the Siemens Somatom Definition Flash 128 slice dual-source CT scanner (Siemens Corp., Erlangen, Germany). Scan protocols included acquisition of the liver using the following parameters: 120 kVp, 200–300 m based on current modulation, collimated slice thickness 0.6 mm, pitch 0.6. The contrast medium (iohexol [trade name Omnipaque^®^] 350 mg/mL, GE Healthcare, Milwaukee, WI, USA) was injected at a rate of 3.3 mL/second through an 18-gauge plastic IV catheter placed in an antecubital vein. The volume of contrast medium delivered varied depending on the body weight of each donor (1.5 mL/kg of body weight), and the total upper limit volume of contrast medium administered was 150 mL. Bolus tracking was used, and the hepatic arteries phase was scanned after a 2-s delay, with the threshold of 180HU in descending thoracic aorta, portal vein phase. Hepatic veins phase was obtained in 25-s and 35-s delays separately after the trigger threshold was reached. Axial images were multiplanar reconstructed on AZE Virtual Place workstation (AZE Corporation, Tokyo, Japan) with 3 mm slice in axial and coronal view and maximum intensity projection (MIP) reconstruction for assessing hepatic vasculature.

### 2.3. MRI Examination

MRI examination was performed using the 1.5 T whole-body scanner (Discovery 450; GE Healthcare, Milwaukee, WI, USA) with multiple routine pulse sequences, including axial T2-weighted SS-SFE (TR/TE: 12,857/87 ms, flip angle: 160°, 5-mm sections, matrix size: 320 × 320) and the non-CE MRA was performed by IFIR pulse sequence; the optimal inversion time (TI) of IFIR was 1400 milliseconds for the arteries, 1800 milliseconds for PVs, and 1800 milliseconds for IVCs, as previously described [18,19]. All liver donor candidates were in supine position with normal respiratory trigger, and 12-channel Body Array Coils (1.5 T Signa HDxt MR System) were prepared. To delineate the vessel of interest, the proper position of selective inversion pulse was tagged to suppress liver, descending aorta, hepatic vein, and portal veins inflow signals. The static tissue signal was acquired by 3D-balanced steady-state free precession (bSSFP) acquisition with chemical fat suppression. The imaging area was centered in the liver hilum covering the whole liver region. The typical scanning parameters were repetition time 3.7 ms, echo time 1.9 ms, flip angle: 55°, blood suppression inversion time 1400 ms for arteries and 1800 ms for portal veins and IVC, matrix 192 × 320, field of view 36 × 36 cm^2^, slice thickness 2.0 mm, slice number 58, readout bandwidth 125.00 kHz, and number of excitations 0.69. Parallel imaging was used with acceleration factor phase 2. After completing the non-CE MRA, all candidates received gadolinium-based contrast medium (Magnevist^®^, BAYER Radiology, Pittsburgh, PA, USA) injection (0.1 mmol/kg, total 10–15 mL), and then MR cholangiography and CMRA were performed using Coronal 3D MRCP (TR/TE: 3000/505 ms, flip angle: 90°, matrix size: 320 × 320) and breath-hold 3D spoiled gradient-echo imaging (TR/TE, 3.6/1.42, flip angle:30°; field of view, 40 × 36 cm; bandwidth, 32 kHz; matrix size, 288 × 192).

### 2.4. Image Interpretation

All hepatic arteries, portal veins, and hepatic veins anatomical types were assessed in CTA. The non-CE MRA vascular depictions were evaluated using subjective visualized image quality [5], containing four levels based on vascular imaging contrast as follows: background imaging as one point for no signal or completely unreadable, two points for visible vascular signal but unable to measure caliber, three points for moderate vessel signal but the vascular wall margin was blurred and not easily differentiated from the background, and four points for optimal vascular signal and clear margin differentiated from the background. All vessels were assessed by two independent radiologists on picture and archiving communication systems (GE Healthcare, Barrington, IL, USA). The differences were then discussed by the two staff members (Ou HY, Liao CC, specialized in gastrointestinal and hepatobiliary imaging for 15 years and 6 years, respectively). All visualized scores were presented as mean ± standard deviation, per the Streitparth et al. method [20].

Arterial anatomy was recorded according to Michel’s classification [21] by identifying the origins of the common, right, and left hepatic arteries and the presence of any accessory hepatic arteries. (Figure 1).

The portal vein anatomy was recorded according to Cheng’s classification [22]. Hepatic veins are interpreted by their drainage territory and the presence of inferior right hepatic vein (IRHV). Interpretation and recognition of all vascular anatomy types were performed in non-CE MRA and CMRA and then compared with CTA imaging as the standard image.

### 2.5. Imaging Analysis

The quantification of vascular signals was also performed by operator-defined region-of-interest (ROI) measurements, which were assessed by two independent radiologists on picture and archiving communication systems (GE Healthcare, Barrington, IL, USA). There were three ROIs drawn in the hepatic venous branches (right, middle, and left), respectively, the ROIs were placed 1 cm proximal to the IVC and drawn as large as possible. There were two to three ROIs placed in primary hepatic arteries (right, left, or accessory artery) and three ROIs in the portal vein (main trunk, right, and left). There were two to four ROIs drawn in intrahepatic arteries branches either from the right hepatic artery (RHA) or left hepatic artery (LHA), while the first and second branches of single arteries were designated RHA-2, RHA-3, LHA-2, and LHA-3. In addition, there were three ROIs drawn in another intrahepatic vascular anatomy essential for donor hepatectomy, including segment IV hepatic artery (S4HA), segment VIII hepatic vein (S8HV), and inferior right hepatic vein (IRHV), which were also identified. ROIs in the liver parenchyma were in a homogenous portion of the liver devoid of vessels and prominent artifacts. Background noise was also recorded as a control. Mean signal intensity (SI) values were adopted. Relative SNR and contrast-to-noise ratio (CNR) were calculated using the following equations: SNR = mean SI values in vascular ROI divided by standard deviation (SD) of air noise and CNR = (mean SI values in vascular ROI–mean SI values in liver parenchyma/SD of air noise) [23,24].

### 2.6. Statistical Analysis

Statistical analysis was performed using commercially available software (SPSS version 22.0, IBM SPSS, Chicago, IL, USA). Interobserver agreement was assessed using Cohen’s kappa correlation coefficient categorized as poor (<0.20), fair (0.20–0.39), moderate (0.40–0.59), substantial (0.60–0.79), and excellent (≥0.80). The calculated CNR of hepatic vessels were presented as mean ± standard deviation; an independent *t*-test was used to compare continuous variables between the non-CE MRA and CMRA group. A *p*-value less than 0.05 was considered to indicate a statistically significant difference.

## 3. Results

A total of 43 potential liver donor candidates received a complete pre-transplantation evaluation, including blood tests, sonography, CTA, and MRI; none of our enrolled donor candidates had complications such as allergy or claustrophobia during MR examination. Twenty-six donors received living donor liver transplantation, with no significant vascular complication after the procedure. All hepatic vasculatures were clearly delineated in the CTA study, including the origin of S4HA, either in the multiplanar reconstructed image or MIP reconstructed image.

The average visualized score (Table 1) of the portal vein was 3.5, and the hepatic vein was 3.4, which makes them the most clearly observed vessels in non-CE MRA. The proximal artery segments PHA, RHA, and LHA had an average score of 3.2 to 3.5, but the first and second branches of hepatic arteries had relatively low visualized scores: 2.3 in RHA-2 and 1.5 in RHA-3, and 1.6 in LHA-2 and 1.2 in LHA-3. Segment VIII hepatic vein (S8HV) had the highest score (3.3) in essential vascular anatomy, IRHV had the median score (3.1), and S4HA had the lowest score (1.7). Interobserver evaluation of non-CE MRA (Table 1) had an almost perfect agreement in LHA, PHA, right hepatic vein, main portal vein, and its major branches (κ > 0.8), with substantial (κ = 0.60 to 0.79) in RHA, S8HV, S4HA, IRHV, middle hepatic vein, and left hepatic veins.

Five types of hepatic artery anatomy were identified based on Michel’s classification using CTA as a standard reference image (Table 2). There were 35 type I donors (normal anatomy), for which non-CE MRA identified 29 out of 35 donors (83%) and CMRA identified 34 out of 35 donors (97%); one type III donor (replaced RHA originating from the superior mesenteric artery), identified by both non-CE MRA and CMRA; five type V donors (accessory LHA originating from the left gastric artery), with four out of five donors (80%) identified by non-CE-MRA identified and five donors (100%) identified by CMRA; one type VI donor (accessory RHA originating from the superior mesenteric artery), identified by both non-CE MRA and CMRA; and one type VIII donor (replaced RHA and accessory LHA, or replaced LHA and accessory RHA), no common hepatic artery (CHA) absence was identified in our studied donor, although several suboptimal image qualities in either CRMA or non-CE MRA, all hepatic artery types were identified by MRI studies.

Three types of portal vein anatomy were identified based on Cheng’s classification using CT angiography as the standard reference image (Table 2). Among 40 type I portal vein (classic ramification) donors, non-CE MRA identified 39 out of 40 donors (99%), and CMRA identified 39 out of 40 donors (99%). One donor candidate had suboptimal portal vein image quality in both CMRA and non-CE MRA; one type II portal vein (trifurcation type) donor and two type III (independent right posterior portal vein) donors were identified by both non-CE MRA and CMRA, which were able to identify all type II and III donors (100%).

The tributary of the hepatic vein includes the single right hepatic vein and separate or common trunks of middle and left hepatic veins (Table 2). All studied donors had a common trunk of middle and left hepatic veins, of which non-CE MRA identified 41 out of 43 donors (95%), CMRA identified 41 out of 43 donors (95%), and 2 donors had suboptimal middle and left hepatic veins depicted in CMRA and non-CE MRA due to respiration motional artifacts and inappropriate venous phase scans. Both CMRA and non-CE MRA identified the tributary of right hepatic veins for all 43 donors with ideal image quality.

Several intrahepatic vascular anatomies are essential for donor hepatectomy, and non-CE MRA identified 22 (51%) S4HA, while CMRA identified 37 (86%) S4HA in all donors. When reviewing either CMRA or non-CE MRA images, S4HA was identified in 40 (93%) donors. Altogether 20 donors had IRHV, of which non-CE MRA identified 16 (80%) IRHV and CMRA identified 15 (75%) IRHV, while IRHV in 19 (95%) donors were identified by either CMRA or non-CE MRA. Non-CE MRA identified S8HV in all donors (100%), but CMRA identified S8HV in 40 donors (93%).

The quantification of each vessel’s signal intensity (Table 3) is based on CNR. The CNRs of the portal vein and hepatic vein were higher in non-CE MRA than in CMRA. The CNR of the main portal vein (MPV) was 11.00 in non-CE MRA and 3.02 in CMRA (*p* < 0.05); for right portal vein (RPV) was 14.27 in non-CE MRA and 3.61 in CMRA (*p* < 0.05); for left portal vein (LPV) was 10.87 in non-CE MRA and 3.75 in CMRA (*p* < 0.05). The CNR of the middle hepatic vein (MHV) was 11.28 in non-CE MRA and 2.90 in CMRA (*p* < 0.05); for right hepatic vein (RHV) was 11.94 in non-CE MRA and 2.66 in CMRA (*p* < 0.05); for left hepatic vein (LHV) was 9.35 in non-CE MRA and 2.01 in CMRA (*p* < 0.05). The CNR of the proper hepatic artery (PHA) was 9.50 in non-CE MRA and 12.76 in CMRA (*p* < 0.05); for RHA was 8.66 in non-CE MRA and 8.99 in CMRA (*p* = 0.90); for LHA was 5.25 in non-CE MRA and 6.75 in CMRA (*p* = 0.13). No significant differences were found in the CNR of right and left hepatic arteries between non-CE MRA and CMRA images.

## 4. Discussion

Results of the present study show that the pretransplant vascular anatomy can be well identified using combined non-CE MRA and CMRA of the hepatic arteries (100%), PVs (98%), and hepatic vein (100%), and results compare favorably with CTA and correlate well with the intraoperative findings in all donors, achieving a 100% diagnostic accuracy rate for pretransplant vascular anatomy. The main feature of IFIR in non-CE MRA is the placement of the selective inversion pulse to saturate the background signal, which then selectively acquires the vessel of interest inflow information [15]. Selectively visualizing the specific vascular structure has the advantage of decreasing the overlapping of unwanted vessels, which allows us to assess the vascular pattern more clearly (Figure 2). Magnetic resonance cholangiopancreatography (MRCP) includes mandatory steps for the evaluation of potential donors in LDLT. Combined non-CE MRA and CMRA demonstrate comparable diagnostic ability with CTA and provides enhanced biliary anatomical information that optimizes donor safety.

CTA is the ideal reference for standard imaging by which to evaluate hepatic vascular structure and liver volume. The present study has shown the optimal diagnostic performance of non-CE MRA and CMRA with results comparable to those of standard CTA. In particular, S4HA origin reorganization is important for donor safety in LDLT when planning extended right liver graft [25]. Shimada et al. [26] demonstrated a relatively low visualization quality of S4HA compared with RHA and LHA in CMRA, possibly due to its relatively small caliber and blood flow signal. The visualized score of S4HA in non-CE MRA had similar low-level scores. To improve the contrast of S4HA to liver parenchyma, a specific inversion time allowing a higher S4HA signal, and proper suppression of the liver parenchyma signal requires further investigation. In the present study, non-CE MRA improved the visualization rate of S4HA from 86% to 93%.

The absence of CHA is another important donor safety issue during donor hepatectomy. It is defined as CHA that does not originate from the celiac trunk but from the SMA or abdominal aorta, the replaced RHA from SMA or celiac trunk, and the replaced LHA from LGA or aberrant gastroduodenal artery (GDA) according to Huang’s classification. Huang et al. [27] reported portal vein and hepatic artery injury during lymph node dissection in laparoscopic surgery for gastric cancer. Cirocchi et al. [28] published one meta-analysis discussing that the overall pooled prevalence estimate (PPE) of an absent CHA was 3.1% and 3.0% for participants who underwent angio-CT evaluation. Although no donor candidate of our studied group presented an absence of CHA, according to our observation, the origin of CHA in this study has optimal signal intensity either in non-CE MRA or CMRA because CHA has a larger caliber and higher signal intensity. During donor hepatectomy in our institution, surgeons routinely performed test clamping of the hepatic artery and checked liver blood flow by intraoperative doppler sonography to minimize the risk of misidentification of all kinds of anatomy variation. Recently, laparoscopic donor hepatectomy has been performed in several studies [29,30,31]. Identifying CHA origin is crucial in performing such a procedure.

The highest priority in LDLT is to obtain a detailed preoperative evaluation to minimize morbidity for both donor and recipient. This is the most important concern, especially for healthy donor candidates, to reduce as much iatrogenic risk as possible. Routine CT angiography and MR angiography require iodine-based contrast medium and gadolinium-based contrast medium. Although gadolinium-based contrast medium has lower allergy rates than iodine-based contrast medium (0.121% and 0.73%) [8,9], it may still cause nephrogenic systemic fibrosis in patients with renal function impairment. Since the organ functions of all donor candidates are supposed to be healthy, nephrogenic systemic fibrosis should not be a threat; however, intracranial gadolinium deposition after contrast-enhanced MRI study has been reported as recently as 2017 [10]. Although the outcome and safety of gadolinium deposition remain to be determined, it is obvious that the less contrast medium used on the donor, the less risk is involved.

When vessels were quantified by acquiring vascular signal intensity and calculating the CNR, the average CNR of the portal vein and hepatic vein in non-CE MRA was significantly higher than that in CMRA, in which the portal venous phase and hepatic venous phase need imaging time delay, while contrast medium flows into both the liver parenchyma and vessel, decreasing the contrast of the vessel compared with the surrounding liver parenchyma. This accounts for the high identification rate of the portal venous system and hepatic venous system in non-CE MRA. No significant differences were shown in the CNR of hepatic arteries between non-CE MRA and CMRA, but the CNR of distal artery branches in non-CE MRA was less than that in CMRA. To identify the anatomic type of the hepatic artery, non-CE MRA has optimal diagnostic ability of the proximal segment of the hepatic artery, which provides sufficient information about hepatic artery type. However, CMRA has better diagnostic ability in identifying S4HA than non-CE MRA. In the present study, the hepatic artery images of two donors had poor-quality CMRA images due to scan delay time errors after contrast medium injection; however, the hepatic artery images had optimal quality in non-CE MRA for identifying the hepatic artery pattern. Since the beneficial characteristic of non-CE MRA is its freedom from contrast medium injection, it can be performed repeatedly. If a donor has suboptimal imaging quality in CTA and CMRA images for diagnosis, non-CE MRA is the ideal alternative of acquiring vascular images without contrast medium administration and repeating the studies until the satisfying image is obtained (Figure 3).

The limitations of non-CE MRA technique are flow dependence, improper shimming of magnetic field, and lack of cooperation with respiration gating, which may compromise image quality [32]. Another limitation is that although the liver background signal is saturated in IFIR scans, the gallbladder and intrahepatic bile duct may also be delineated due to their high intrinsic T2/T1 signal intensity in bSSFP sequences [14]. Proper inversion time was adjusted in the present study to null the bile duct signal as much as possible; however, it does cause interference of image interpretation. Additionally, performing non-CE MRA extends the examination time—about 10 minutes. Using two-dimensional parallel imaging and short tau inversion recovery may shorten the scan time [26]. In a situation when both non-CE MRA and CMRA fail to obtain optimal images, non-CE MRA cannot be repeated immediately within the same examination if CMRA has already been performed because the use of contrast medium changes the liver parenchyma T1 shortening time, and the inversion time in scan protocol may not be optimal, resulting in poor signal intensity of vascular flow.

The features of MRI in LDLT preoperative evaluation has multiple informative imaging package, including liver parenchymal lesion detection, fatty liver percentage calculation, vascular anatomy evaluation, and cholangiography [33]. Liver volumetry is a crucial component of liver donor candidate evaluation. An accurate liver volumetry can assure donor safety and reduce the risk of graft failure [33]. Liver volume is calculated primarily via liver segmentation of CT or MR images [34]. Contrast-enhanced MR images have higher signal intensity in the liver parenchyma [35]. Besides liver parenchyma, slice thickness also affects liver volume calculation, with thinner slices having more accurate results in liver volume [36] but prolonging the processing time of manual liver segmentation and MR examination [37]. Thus, these are the limitation of MR liver volumetry to replace CT study [38].

Recent studies have reported comparable results of CMRA for preoperative evaluation of hepatic vascular anatomy in living liver donors [39,40], while only a few published studies have focused on the use of non-CE MRA for hepatic portography and recipients of liver transplantation [19,41,42]. This is the first study that has added non-CE MRA in LDLT preoperative evaluation and revealed promising diagnostic results comparable to those of CTA.

## 5. Conclusions

The superiorities of non-CE MRA compared with CMRA are higher vascular signal intensities in hepatic veins, S8HV, IRHV, and portal veins, with comparable results as in CTA and CMRA. It has limited diagnostic ability of S4HA, but optimal results in interpreting hepatic arteries pattern including replaced RHA, replaced LHA, accessory RHA, and accessory LHA. Although the absence of CHA is not found in our study group, CHA still has identifiable signal intensity based on our study. Non-CE MRA decreases contrast medium usage in donor examinations but requires extra examination time. Nevertheless, it can increase the diagnostic performance of MRA studies and can also be an additional donor examination without contrast medium usage and extra radiation dose when CMRA and CTA fail to provide sufficient diagnostic information of hepatic vessels.

## Figures and Tables

**Figure 1 diagnostics-12-00498-f001:**
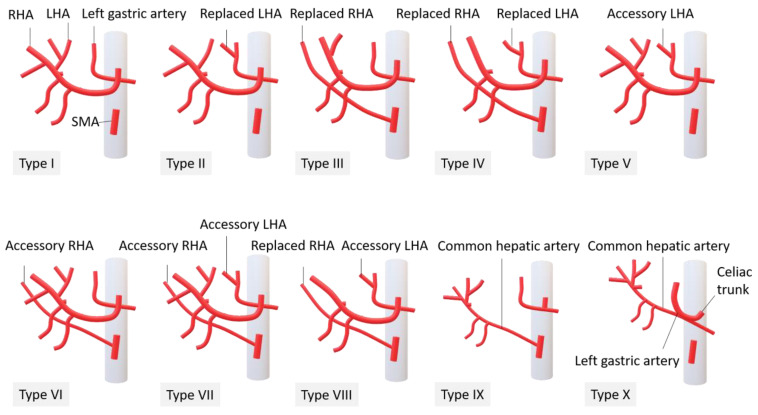
Michel’s classification of hepatic arteries, manually drawing courtesy of Miss Wan-Ching Chang, research assistant, Department of Diagnostic Radiology, Kaohsiung Chang Gung Memorial Hospital.

**Figure 2 diagnostics-12-00498-f002:**
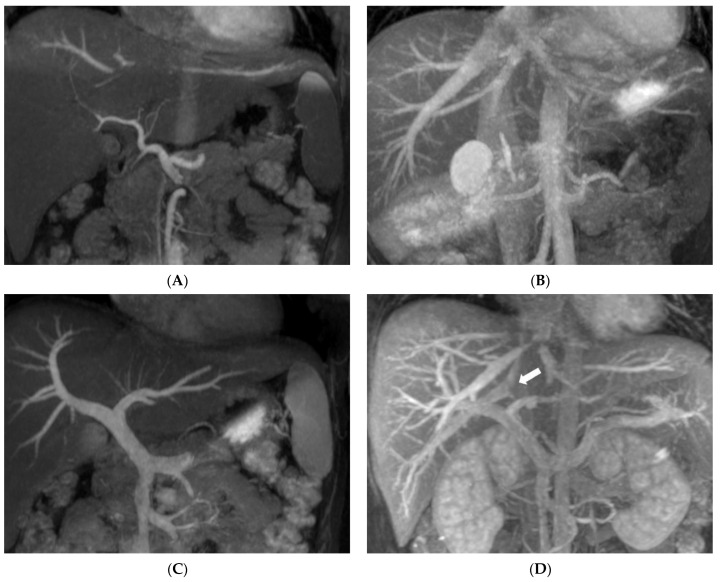
Selective visualization of hepatic vasculature in non-CE MRA compared with CMRA: (**A**) Selective depiction of hepatic artery of a 24-year-old female donor. Partial hepatic vein signal in the superior part of the liver due to non-saturated region, but without obscuring hepatic artery. (**B**) Clear depiction of hepatic veins of a 32-year-old male donor without portal vein overlapping; high signal of gallbladder was also depicted but does not interfere with the interpretation of hepatic veins. (**C**) A 24-year-old female donor’s portal veins are also clearly depicted. (**D**) CMRA of hepatic veins in the same donor as in (**C**), with portal vein overlapping and one IRHV (white arrow) crossing over the right portal vein.

**Figure 3 diagnostics-12-00498-f003:**
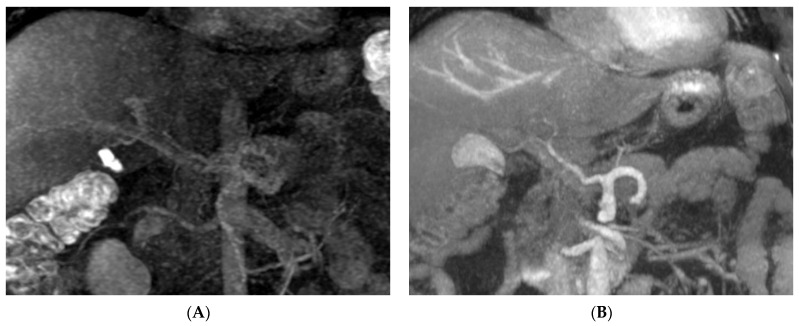
Non-CE MRA is an alternative way of acquiring vascular image. A 45-year-old male donor candidate has suboptimal hepatic artery image in CMRA (**A**). The non-CE MRA (**B**) provides diagnosable delineation of hepatic artery pattern.

**Table 1 diagnostics-12-00498-t001:** Mean image quality scores and interobserver agreement.

Visualized Score	Score (Mean ± SD)	Kappa Value
MPV	3.5 ± 0.7	0.8
RPV	3.5 ± 0.7	0.9
LPV	3.5 ± 0.7	0.8
PHA	3.5 ± 0.7	0.8
RHA	3.2 ± 1.0	0.7
RHA-2	2.3 ± 1.2	0.5
RHA-3	1.5 ± 0.8	0.6
LHA	3.0 ± 1.0	0.9
LHA-2	1.6 ± 0.9	0.8
LHA-3	1.2 ± 0.6	0.4
MHV	3.4 ± 0.8	0.7
RHV	3.5 ± 0.6	0.9
LHV	3.3 ± 0.9	0.7
S4 HA	1.7 ± 0.9	0.7
S8 HV	3.3 ± 0.8	0.7
IRHV	3.1 ± 1.2	0.7

**Table 2 diagnostics-12-00498-t002:** Diagnostic performance of non-CE MRA and CMRA.

Michel’s Classification of Hepatic Arteries	CTA	Non-CE MRA	CMRA	CMRA + Non-CE MRA	Pearson Chi-Square *p*-Value
I	35	29/35 (83%)	34/35 (97%)	35/35 (100%)	
III	1	1/1 (100%)	1/1 (100%)	1/1 (100%)	
V	5	4/5 (80%)	5/5 (100%)	5/5 (100%)	
VI	1	1/1 (100%)	1/1 (100%)	1/1 (100%)	
VIII	1	1/1 (100%)	0/1 (0%)	1/1 (100%)	
Total	43	36/43 (84%)	41/43 (95%)	43/43 (100%)	<0.01
**Portal vein ramification**					
Type I	40	39/40 (99%)	39/40 (99%)	39/40 (99%)	
Type II	1	1/1 (100%)	1/1 (100%)	1/1 (100%)	
Type III	2	2/2 (100%)	2/2 (100%)	2/2 (100%)	
Total	43	42/43 (98%)	42/43 (98%)	42/43 (98%)	
**Hepatic vein tributary**					
Common trunk of MHV and LHV	43	41/43 (95%)	41/43 (95%)	43/43 (100%)	
Single RHV	43	43/43(100%)	43/43 (100%)	43/43(100%)	
Total	46	84/86 (98%)	84/86 (98%)	86/86 (100%)	
**Essential vascular anatomy**					
S4HA from either RHA or LHA	43	22/43 (51%)	37/43 (86%)	40/43 (93%)	<0.01
IRHV	20	16/20 (80%)	15/20 (75%)	19/20 (95%)	<0.01
S8HV	43	43/43 (100%)	40/43 (93%)	43/43 (100%)	
Total	106	81/106 (75%)	92/106 (87%)	102/106 (96%)	

**Table 3 diagnostics-12-00498-t003:** Quantification of vessel signal intensity based on CNR.

Group	Mean ± SD	*p*-Value
Non-CE MPV-CNR	11.00 ± 7.12	<0.01
C-MPV-CNR	3.02 ± 2.42	
Non-CE RPV-CNR	14.27 ± 14.04	<0.01
C-RPV-CNR	3.61 ± 2.53	
Non-CE LPV-CNR	10.87 ± 6.51	<0.01
C-LPV-CNR	3.75 ± 2.49	
Non-CE PHA-CNR	9.50 ± 5.43	<0.01
C-PHA-CNR	12.76 ± 6.43	
Non-CE RHA-CNR	8.66 ± 15.21	0.90
C-RHA-CNR	8.99 ± 6.97	
Non-CE LHA-CNR	5.25 ± 4.44	0.13
C-LHA-CNR	6.75 ± 4.94	
Non-CE MHV-CNR	11.28 ± 5.81	<0.01
C-MHV-CNR	2.90 ± 2.24	
Non-CE RHV-CNR	11.94 ± 6.23	<0.01
C-RHV-CNR	2.66 ± 2.12	
Non-CE LHV-CNR	9.35 ± 5.03	<0.01
C-LHV-CNR	2.01 ± 2.14	

## Data Availability

Restrictions apply to the availability of these data. Data was obtained from Kaohsiung Chang Gung Memorial Hospital and are available C.-C.L., H.-Y.O., C.-C.L. and Y.-F.C. with the permission of Kaohsiung Chang Gung Memorial Hospital, Taiwan.

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
