# Peer review of "Non-Contrast-Enhanced and Contrast-Enhanced Magnetic Resonance Angiography in Living Donor Liver Vascular Anatomy"

_diagnostics, 2022, doi:10.3390/diagnostics12020498_

Round 1

Reviewer 1 Report

Dear Editor and Authors

Thank very much for your request to revise the manuscript of Liao et coll. “Non-Contrast-Enhanced and Contrast Enhanced Magnetic Resonance Angiography in Living Donor Liver Vascular Anatomy”. It is a pleasure for me read the manuscript reported.

The manuscript is very interesting and only few revisions are needed. The Authors used the classification of Michels as following: Type I: normal pattern; type II: a replaced LHA from the left gastric artery; type III: a replaced RHA from the SMA; type IV: replaced RHA and LHA; type V: an accessory LHA; type VI: an accessory RHA; type VII: accessory RHA and LHA; type VIII: a replaced RHA or LHA with other hepatic artery being an accessory one; type IX: the hepatic trunk as a branch of the SMA; and type X: the CHA from the left gastric artery. Nowadays other Authors used new classifications as Huang (Huang, Chang-Ming et al. “Application Value of a 6-Type Classification System for Common Hepatic Artery Absence During Laparoscopic Radical Resections for Gastric Cancer: A Large-Scale Single-Center Study.” Medicine vol. 94,32 (2015): e1280. doi:10.1097/MD.0000000000001280).

The Absence of Common Hepatic Artery (CHA) is a critical point in liver transplantation, the data of CHA are not clear reported in the manuscript although it is possible perform a sum of the Michels’ type that include the CHA absence.

In literature, this data is very heterogeneous in the largest analysis performed from Cirocchi et. Coll. on fifty-four articles were included in the review (26,250 participants). In this study, the overall pooled
prevalence estimate (PPE) of an absent CHA was 3.1%. Of those participants who underwent preoperative radiological evaluation, the overall PPE of an absent CHA was 3.8% for subjects who were evaluated via angiography and 3.0% for participants who underwent angio-CT evaluation. The overall PPE of an absent CHA was 3.9% in cadavers and 3.2% in participants evaluated surgically.

I suggest reporting the your data about the CHA’ absence and discuss this result in the discussion in comparation with the Cirocchi' article.

Cirocchi R, D'Andrea V, Lauro A, Renzi C, Henry BM, Tomaszewski KA, Rende M, Lancia M, Carlini L, Gioia S, Randolph J. The absence of the common hepatic artery and its implications for surgical practice: Results of a systematic review and meta-analysis. Surgeon. 2019 Jun;17(3):172-185. 

Author Response

Dear Editors and reviewers, Diagnostics,

  Thanks for your review of our manuscript (ID: diagnostics-1582527) entitled Non-Contrast-Enhanced and Contrast-Enhanced Magnetic Resonance Angiography in Living Donor Liver Vascular Anatomy, which has been recently returned along with the reviewer’s comments and suggestions.

Point 1: The absence of Common Hepatic Artery (CHA) is a critical point in liver transplantation, the data of CHA are not clearly reported in the manuscript although it is possible to perform a sum of the Michels’ type that includes the CHA absence.

Author Response of point 1:

We appreciate the reviewer for bringing this to our attention. There is a similar finding of CHA absence as Michel’s type IX and X which are not identified in our study group after reviewing images.

Author action of point 1:

We will add this result regarding Michel’s classification and Huang’s classification into the revised manuscript. Page 6, line 214-215

Point 2: I suggest reporting your data about the CHA’ absence and discussing this result in the discussion in comparison with the Cirocchi article.

Author Response of point 2:

Thanks for your suggestion. In this revised manuscript, we have added a new subsection discussing Cirocchi’s study and several new reference articles.

Author action of point 2:

The added paragraph is shown below: page 9, line 289-304

“The absence of CHA is another important donor safety issue during donor hepatectomy. It is defined as CHA that does not originate from the celiac trunk but SMA or abdominal aorta, the replaced RHA from SMA or celiac trunk, and the replaced LHA from LGA or aberrant gastroduodenal artery (GDA) according to Huang’s classification. Huang et al. reported portal vein and hepatic artery injury during lymph node dissection in laparoscopic surgery for gastric cancer. Cirocchi et al. has published one meta-analysis discussing the overall pooled prevalence estimate (PPE) of an absent CHA was 3.1% and 3.0% for participants who underwent angio-CT evaluation. Although no donor candidate of our studied group presented an absence of CHA, according to our observation, the origin of CHA in this study has optimal signal intensity either in non-CE MRA or CMRA because CHA has a larger caliber and higher signal intensity. During donor hepatectomy in our institution, surgeons routinely performed test clamping of the hepatic artery and checked liver blood flow by intraoperative doppler sonography to minimize the risk of misidentification of all kinds of anatomy variation. Recently, laparoscopic donor hepatectomy has been increasing in several studies. CHA origin is crucial in performing such a procedure.”

These three articles will be added to the reference list, along with Huang et al. and Cirocchi et al.’s articles.

  1. Hasegawa Y, Nitta H, Takahara T, Katagiri H, Kanno S, Sasaki A. Pure laparoscopic living donor hepatectomy using the Glissonean pedicle approach (with video). Surg Endosc. 2019 Aug;33(8):2704-2709. doi: 10.1007/s00464-019-06818-7.
  2. Dokmak S, Cauchy F, Sepulveda A, Choinier PM, Dondéro F, Aussilhou B, Hego C, Chopinet S, Infantes P, Weiss E, Francoz C, Sauvanet A, Paugam-Burtz C, Durand F, Soubrane O. Laparoscopic Liver Transplantation: Dream or Reality? The First Step With Laparoscopic Explant Hepatectomy. Ann Surg. 2020 Dec;272(6):889-893. doi: 10.1097/SLA.0000000000003751.
  3. Peng Y, Li B, Xu H, Chen K, Wei Y, Liu F. Pure Laparoscopic Versus Open Approach for Living Donor Right Hepatectomy: A Systematic Review and Meta-Analysis. J Laparoendosc Adv Surg Tech A. 2021 Nov 25. doi: 10.1089/lap.2021.0583. Epub ahead of print. PMID: 34842460.

We have studied your comments carefully and have made amendments which are marked in red in the manuscript. We would like to express our gratitude to you for the comments and suggestions regarding our manuscript. It will be highly appreciated if the paper in its present revised form is accepted for publication in your esteemed journal. If there are any further comments or necessary adjustments, please do not hesitate to let us know.

Thank you and best regards.

Dr. Chien Chang Liao
Department of Radiology
Kaohsiung Chang-Geng Memorial Hospital, Taiwan.
+886 975 056 724
[email protected]

Reviewer 2 Report

Actually, MRI might be "all-in-one" imaging package including liver parenchymal lesion detection, fatty liver percentage calculation, vascular anatomy evaluation and cholangiography.

As authors mentioned, the highest priority in LDLT is to obtain a detailed preoperative evaluation to minimize morbidity for both donor and recipient. 

In donor’s surgery, the precise spacial anatomy of all vessels including HA, PV, and HV should be evaluated before operation.

In addition, the precise estimation of the remnant liver volume are important for donor’s safety.

I’m wondering wether this modality (Non-CE MRA) have enough ability to undergo donor’s surgery safely.

Major:

Authors have to reveal the superiorities comparing to conventional modalities (CTA and CMRA) regarding the spacial anatomy of all vessels and the estimation of the remnant liver volume.

Minor:

Authors have to clarify Michel’s classification using schema.

Author Response

Dear Editors and reviewers, Diagnostics,

  Thanks for your review of our manuscript (ID: diagnostics-1582527) entitled Non-Contrast-Enhanced and Contrast-Enhanced Magnetic Resonance Angiography in Living Donor Liver Vascular Anatomy, which has been recently returned along with the reviewer’s comments and suggestions.

Point 1: Authors have to reveal the superiorities compared to conventional modalities (CTA and CMRA) regarding the special anatomy of all vessels and the estimation of the remnant liver volume.

Response of point 1-1, the superiorities of non-CE MRA:

Thanks for your suggestion; the discussion section addressed the identification rate of combined non-CE MRA, and CMRA has 98-100% in interpreting hepatic arteries, veins, and portal veins. In the conclusion section, we will add the following paragraph regarding the superiorities of non-CE MRA, to emphasize our findings.

“The superiorities of non-CE MRA compared to CMRA are higher vascular signal intensities in hepatic veins, S8HV, IRHV, and portal veins, with comparable results as in CTA and CMRA. It has limited diagnostic ability of S4HA, but optimal results in interpreting hepatic arteries including replaced RHA, replaced LHA, accessory RHA, and accessory LHA. Non-CE MRA decreases contrast medium usage in donor examinations but requires extra examination time. Still, it can increase the diagnostic performance of MRA studies and can also be an additional donor examination without contrast medium usage when CMRA and CTA fail to provide sufficient diagnostic information of hepatic vessels.”

Response of point 1-2, the estimation of the remnant liver of the donor

We will add the following paragraph regarding MR liver volumetry, page 11, line 356-363

“Liver volumetry is a crucial component of liver donor candidate evaluation. An accurate liver volumetry can assure donor safety and reduce the risk of graft failure. Liver volume is calculated primarily via liver segmentation of CT or MR images. Contrast-enhanced MR images have higher signal intensity in the liver parenchyma. Besides liver parenchyma, slice thickness also affects liver volume calculation, with thinner slices having more accurate results in liver volume and prolonging the processing time of manual liver segmentation and MR examination. Thus, these are the limitation of MR liver volumetry to replace CT study.”

Point 2: Authors have to clarify Michel’s classification using schema.

Response of point 2

We appreciate and completely agree with the reviewer’s suggestion, we will add a schema to clarify Michel’s classification of hepatic arteries

Figure 1 Michel’s classification of hepatic arteries, manually drawing courtesy of Miss Wan-Ching Chang, research Assistant, Department of Diagnostic Radiology, Kaohsiung Chang Gung Memorial Hospital

We have studied your comments carefully and have made amendments which are marked in red in the manuscript. We have tried our best to revise our manuscript in accordance with the comments.

We would like to express our gratitude to you for the comments and suggestions regarding our manuscript. It will be highly appreciated if the paper in its present revised form is accepted for publication in your esteemed journal. If there are any further comments or necessary adjustments, please do not hesitate to let us know.

Thank you and best regards.

Best regards,
Dr. Chien Chang Liao
Department of Radiology
Kaohsiung Chang-Geng Memorial Hospital, Taiwan.
+886 975 056 724
[email protected]

Reviewer 3 Report

Comments to the Author

Title: Non-Contrast-Enhanced and Contrast Enhanced Magnetic Resonance Angiography in Living Donor Liver Vascular Anatomy

Authors: Chien-Chang Liao MD, Meng-Hsiang Chen MD, Chun-Yen Yu MD, Leung-Chit Leo Tsang MD, Chao-Long Chen MD, Hsien-Wen Hsu MD, Wei-Xiong Lim MD, Yi‐Hsuan Chuang MD, Po-Hsun Huang MD, 1u‐Fan Cheng MD and Hsin‐You Ou MD

à The paper explores the diagnostic efficacy of non-contrast-enhanced magnetic resonance angiography, contrast enhanced magnetic resonance angiography and computed tomography angiography in the preoperative evaluation of living liver donors.

à The study reported in the manuscript is interesting and well presented. I would endorse the publication of the present manuscript after carefully addressing the comments hereafter.

General: The manuscript is well written and the sections are adequate.  

In the following: MA=Major comment, MI = Minor comment, OP = Optional Comment

 (MI) P1, Abstract, L21: There is a typo, CMRA à C-MRA

(OP) P1, Introduction, L44: A reference to support this statement should be included

(MI) P2, Introduction, L49: In the abstract section you used C-MRA but now you are using CMRA, please unify styles

(OP) P2, Introduction, L52: A reference to support this statement should be included

 (MI) P2, Introduction, L64-71: Please, provide reference supporting these statements and to the works you are mentioning

(MI) P2, Introduction, L72: Describe what IFIR means

(MI) P2, Introduction, L79-81: Please, provide references supporting these claims

(MA) P2, Material and Methods, 88: In the abstract you said 43 donors. Please correct

(OP) P3, Material and Methods, 98: You can add a reference with the datasheet of the scanner so the reader can read more about the specification details of the system

(OP) P3, Material and Methods, 112: You can add a reference with the datasheet of the scanner so the reader can read more about the specification details of the GE system

(MI) P3, Material and Methods, L125: There is a typo, 36x36 cm à36x36 cm2

(OP) P3, Material and Methods: Did you experienced any complication during patient management? It would be useful to add 2-3 lines commenting on this.

(MI) P4, Material and Methods, 149-161: How many ROIs were drawn?

(MI) P4, Material and Methods, 159: Briefly explain what SI values are or add a reference

(OP) P5, Results, 176-184: Some of this information belongs to the method section, consider reformulating this paragraph to make reading more fluid

(MI) P6, Results, Table I: Explain in the corresponding section of the manuscript how the error shown in this table are estimated. Also, all values should present the same number of significative digits

(MI) P6, Results, L237: CNR and Roi have been already defined

(MI) P6, Results, Table 3: CNR has been already defined. As in Table 1 explain how the error were calculated

(MI) P8, Results, L284: There is a typo: TI à T1

(MI) P8, Results, L288: There is a hyperlink associated to morbidity. Why?

(OP) P9, Results, All references to figures: It would be better to write Figure 2B than Figure2B. This is just a suggestion for all references to figures across the manuscript

(OP) P10, Conclusion, L350-353: Elaborate a bit more this section to emphasize on your findings

Author Response

Dear Editors and reviewers, Diagnostics,

  Thanks for your review of our manuscript (ID: diagnostics-1582527) entitled Non-Contrast-Enhanced and Contrast-Enhanced Magnetic Resonance Angiography in Living Donor Liver Vascular Anatomy, which has been recently returned along with the reviewer’s comments and suggestions. The changes and responses are outlined as follows:

(MI) P1, Abstract, L21: There is a typo, CMRA à C-MRA

Response: Thanks for your comment, The C-MRA has been corrected to CMRA.

(OP) P1, Introduction, L44: A reference to support this statement should be included

Author response:

We apologize for not providing proper references in this paragraph.

Author action: The following articles will be added to the reference list,

1.Alonso-Torres A, Fernández-Cuadrado J, Pinilla I, Parrón M, de Vicente E, López-Santamaría M. Multidetector CT in the evaluation of potential living donors for liver transplantation. Radiographics. 2005 Jul-Aug;25(4):1017-30. doi: 10.1148/rg.254045032. PMID: 16009821.

2.Hecht EM, Kambadakone A, Griesemer AD, Fowler KJ, Wang ZJ, Heimbach JK, Fidler JL. Living Donor Liver Transplantation: Overview, Imaging Technique, and Diagnostic Considerations. AJR Am J Roentgenol. 2019 Jul;213(1):54-64. doi: 10.2214/AJR.18.21034. Epub 2019 Apr 11. Erratum in: AJR Am J Roentgenol. 2019 Sep;213(3):723. PMID: 30973783.

(MI) P2, Introduction, L49: In the abstract section you used C-MRA but now you are using CMRA, please unify styles

Response: All “C-MRA” in the manuscript have been unified using “CMRA”.

(OP) P2, Introduction, L52: A reference to support this statement should be included

Author Response: We would like to rephrase the statement to” The advantages of MRI study include that it is free of ionizing radiation and delineates detailed bile duct anatomy.”,

The following articles will be added to the reference list

“Yeh BM, Breiman RS, Taouli B, Qayyum A, Roberts JP, Coakley FV. Biliary tract depiction in living potential liver donors: comparison of conventional MR, mangafodipir trisodium-enhanced excretory MR, and multi-detector row CT cholangiography--initial experience. Radiology. 2004 Mar;230(3):645-51. doi: 10.1148/radiol.2303021775. PMID: 14990830.”

“Kinner S, Steinweg V, Maderwald S, Radtke A, Sotiropoulos G, Forsting M, Schroeder T. Bile duct evaluation of potential living liver donors with Gd-EOB-DTPA enhanced MR cholangiography: Single-dose, double dose or half-dose contrast enhanced imaging. Eur J Radiol. 2014 May;83(5):763-7. doi: 10.1016/j.ejrad.2014.02.012. Epub 2014 Feb 23. PMID: 24637070.”

(MI) P2, Introduction, L64-71: Please, provide reference supporting these statements and to the works you are mentioning

Author response:

We apologize for not providing proper references in this paragraph.

Author action:

The following articles will be added to support the statements in this paragraph

  1. Laub GA. Time-of-flight method of MR angiography. Magn Reson Imaging Clin N Am. 1995 Aug;3(3):391-8. PMID: 7584245.
  2. Masaryk TJ, Laub GA, Modic MT, Ross JS, Haacke EM. Carotid-CNS MR flow imaging. Magn Reson Med. 1990 May;14(2):308-14. doi: 10.1002/mrm.1910140215. PMID: 2345510.
  3. Shimada K, Isoda H, Okada T, Kamae T, Arizono S, Hirokawa Y, Togashi K. Unenhanced MR portography with a half-Fourier fast spin-echo sequence and time-space labeling inversion pulses: preliminary results. AJR Am J Roentgenol. 2009 Jul;193(1):106-12. doi: 10.2214/AJR.08.1626. PMID: 19542401
  4. Shimada, K.; Isoda, H.; Okada, T.; Kamae, T.; Maetani, Y.; Arizono, S.; Hirokawa, Y.; Shibata, T.; Togashi, K. Non-contrast-enhanced MR angiography for selective visualization of the hepatic vein and inferior vena cava with true steady-state free-precession sequence and time-spatial labeling inversion pulses: preliminary results. J Magn Reson Imaging 2009, 29, 474-479, doi:10.1002/jmri.21636.
  5. Miyazaki, M.; Isoda, H. Non-contrast-enhanced MR angiography of the abdomen. Eur J Radiol 2011, 80, 9-23, doi:10.1016/j.ejrad.2011.01.093.

(MI) P2, Introduction, L72: Describe what IFIR means

Author response: IFIR is an abbreviation that stands for inflow-sensitive inversion recovery.

Author action:

We will add this clarification to the manuscript. Page2-line 73.

(MI) P2, Introduction, L79-81: Please, provide references supporting these claims

Author response:

We apologize for not providing proper references in this paragraph.

Author action:

The following articles will be added to support the statements in this paragraph

Schroeder T, Nadalin S, Stattaus J, Debatin JF, Malagó M, Ruehm SG. Potential living liver donors: evaluation with an all-in-one protocol with multi-detector row CT. Radiology. 2002 Aug;224(2):586-91. doi: 10.1148/radiol.2242011340. PMID: 12147860.

(MA) P2, Material and Methods, 88: In the abstract, you said 43 donors. Please correct

Author response: Thanks for your comment, there were 44 donor candidates enrolled in our study, only 43 of them successfully completed the extra non-CE MRA examination, and one donor failed to gain optimal image due to a respiratory gating problem. We mentioned this situation in the result section. We think it would be appropriate to mention it in the material and method section.

Author action:

We will rephrase the description in the abstract and material section to avoid misunderstanding.

Previous description in the abstract:

A total of 43 liver donor candidates who were evaluated for living donor liver transplantation were enrolled.

Revised description:

page1, line26-27

A total of 43 liver donor candidates who were evaluated for living donor liver transplantation completed examinations.

Page2, line 89-91

43 donor candidates successfully completed the extra non-CE MRA examination, and one donor failed to gain optimal image due to a respiratory gating problem.

(OP) P3, Material and Methods, 98: You can add a reference with the datasheet of the scanner so the reader can read more about the specification details of the system

Author response:

Thanks for your suggestion, however, despite we tried to search the detail scanner specification, there was no optimal result regarding both CT and MR scanners in our institution. We consulted the technician of Siemens, the vendor of our CT scanner, their answer was the detailed scanner specifications are usually available in the official operation manual, which is not searchable online, not even on their website. We apologize for not being able to provide proper references of specifications for the Siemens CT scanner we used in this study.

(OP) P3, Material and Methods, 112: You can add a reference with the datasheet of the scanner so the reader can read more about the specification details of the GE system

Author response:

Thanks for your suggestion, however, despite we tried to search the detail scanner specification, there was no optimal result regarding both CT and MR scanners in our institution. We consulted the technician of GE, the vendor of our MRI scanner, their answer was the detailed scanner specifications are usually available in the official operation manual, which is not searchable online, not even on their website. We apologize for not being able to provide proper references of specifications for the GE MRI scanner we used in this study.

(MI) P3, Material and Methods, L125: There is a typo, 36x36 cm à36x36 cm2

Author response:

Thanks for your comment, we will correct it.

(OP) P3, Material and Methods: Did you experienced any complication during patient management? It would be useful to add 2-3 lines commenting on this.

Author response:

Thanks for your comment, actually, there was no complication of our study group during the examination, not even allergy or claustrophobia. A total of 26 donors in our enrolled group received living donor liver transplantation, with no major vascular complication after the procedure.

Author action:

The first paragraph in the result section will be revised as follow

Page5, line 187-188

none of our enrolled donor candidates had complications such as allergy or claustrophobia during MR examination.

Page 5, line 189

with no significant vascular complication after the procedure.

(MI) P4, Material and Methods, 149-161: How many ROIs were drawn?

Author response:

This paragraph will be revised in page4, line 167-176

There were three ROIs drawn in the hepatic venous branches (right, middle, and left), respectively, the ROIs were placed 1 cm proximal to the IVC and drawn as large as possible; There were two to three ROIs was placed in primary hepatic arteries (right, left, or accessory artery) and three ROIs in the portal vein (main trunk, right and left). There were two to four ROIs drawn in intrahepatic arteries branches either from the right hepatic artery (RHA) or left hepatic artery (LHA), while the first and second branch of single arteries were designated RHA-2, RHA-3, LHA-2 and LHA-3. In addition, there were three ROIs drawn in another intrahepatic vascular anatomy essential for donor hepatectomy including segment IV hepatic artery (S4HA), segment VIII hepatic vein (S8HV) and inferior right hepatic vein (IRHV), which were also identified. In summary, there are about 13-14 ROIs drawn in a donor.

(MI) P4, Material and Methods, 159: Briefly explain what SI values are or add a reference

Author response:

Thanks for your suggestion, we will add detail description of the equation and related reference.

Author action:

Revised paragraph in page 4, line 172-176

Mean signal intensity (SI) values were adopted. Relative SNR and contrast-to-noise ratio (CNR) were calculated using the following equations: SNR= mean SI values in vascular ROI divided by standard deviation (SD) of air noise and CNR= (mean SI values in vascular ROI- mean SI values in liver parenchyma/SD of air noise.

Reference:

  1. Magnotta VA, Friedman L; FIRST BIRN. Measurement of Signal-to-Noise and Contrast-to-Noise in the fBIRN Multicenter Imaging Study. J Digit Imaging. 2006 Jun;19(2):140-7. doi: 10.1007/s10278-006-0264-x. PMID: 16598643; PMCID: PMC3045184.
  2. Chiang HJ, Hsu HW, Chen PC, Chiang HW, Huang TL, Chen TY, Chen CL, Cheng YF. Magnetic resonance cholangiography in living donor liver transplantation: comparison of preenhanced and post-gadolinium-enhanced methods. Transplant Proc. 2012 Mar;44(2):324-7. doi: 10.1016/j.transproceed.2011.12.035. PMID: 22410007.

(OP) P5, Results, 176-184: Some of this information belongs to the method section, consider reformulating this paragraph to make reading more fluid

Author response:

Thanks for your suggestion, we will reformulate this paragraph in the image analysis section of material and method.

(MI) P6, Results, Table I: Explain in the corresponding section of the manuscript how the error shown in this table are estimated. Also, all values should present the same number of significative digits

Author response

Thanks for your comment. Although visualized scores are categorical variables, we used a similar method as the Streitparth et al. method (reference will be listed) to calculate these scores as continuous variables to semi-quantize the visibility of vascular image quality be less subjective.

Author response:

We will list the method as the reference in page4, line 148.

We will change all values in table 1 to be presented in the same number of significative digits.

We would like to correct one wrong calculation of IRHV to 3.1±1.2 in the bottom of table 1

We will revise the result section page 5, line 193-198 as follow

The average visualized score (Table 1) of the portal vein was 3.5 and the hepatic vein was 3.4, which makes them the most clearly seen vessels in non-CE MRA. The proximal artery segments PHA, RHA and LHA had average score of 3.2 to 3.5, but the first and second branch of hepatic arteries had relatively low visualized scores, 2.3 in RHA-2 and 1.5 in RHA-3; 1.6 in LHA-2 and 1.2in LHA-3. Segment VIII hepatic vein (S8HV) had the highest score (3.3) in essential vascular anatomy, IRHV had the median score (3.1), and S4HA had the lowest score (1.7).

(MI) P6, Results, L237: CNR and Roi have been already defined

Author response:

Thanks for your suggestion, we will reformulate this paragraph in the image analysis section of material and method.

(MI) P6, Results, Table 3: CNR has been already defined. As in Table 1 explain how the error were calculated

Author response:

The calculated CNR of hepatic vessels were presented as mean ± standard deviation, an independent t test was used to compare continuous variables between the non-CE MRA and CMRA group. A P-value less than 0.05 was considered to indicate a statistically significant difference.

(MI) P8, Results, L284: There is a typo: TI à T1

Author response:

Thanks for your comment, the TI is an abbreviation that stands for inversion time, not T1, we will use the full name to avoid misunderstanding

Author action

Change TI to inversion time. Page 9, line 286.

(MI) P8, Results, L288: There is a hyperlink associated to morbidity. Why?

Author response:

Thanks for your correction, we will remove it since it was a typo.

(OP) P9, Results, All references to figures: It would be better to write Figure 2B than Figure2B. This is just a suggestion for all references to figures across the manuscript

Author response:

Thanks for your suggestion, we will revise the figure annotation with blank.

(OP) P10, Conclusion, L350-353: Elaborate a bit more this section to emphasize on your findings

Author response:

Thanks for your suggestion, we will comprise another reviewer’s suggestion about the absence of CHA, a crucial arterial variation, to emphasize our findings.

Author action:

The following revised paragraph will be in the conclusion section.

“The superiorities of non-CE MRA compared to CMRA are higher vascular signal intensities in hepatic veins, S8HV, IRHV, and portal veins, with comparable results as in CTA and CMRA. It has limited diagnostic ability of S4HA, but optimal results in interpreting hepatic arteries including replaced RHA, replaced LHA, accessory RHA, and accessory LHA. Although the Non-CE MRA decreases contrast medium usage in donor examinations but requires extra examination time. Still, it can increase the diagnostic performance of MRA studies and can also be an additional donor examination without contrast medium usage and extra radiation dose when CMRA and CTA fail to provide sufficient diagnostic information of hepatic vessels.”

We have studied your comments carefully and have made amendments which are marked in red in the manuscript. We have tried our best to revise our manuscript in accordance with the comments.

We would like to express our gratitude to you for the comments and suggestions regarding our manuscript. It will be highly appreciated if the paper in its present revised form is accepted for publication in your esteemed journal. If there are any further comments or necessary adjustments, please do not hesitate to let us know.

Thank you and best regards.

Dr. Chien Chang Liao
Department of Radiology
Kaohsiung Chang-Geng Memorial Hospital, Taiwan.
+886 975 056 724
[email protected]

Round 2

Reviewer 2 Report

This paper has been well revised.